# CROSS-MODAL FLOWS FOR MULTIMODAL GENERATION

## ABSTRACT

Flow Matching and Diffusion models have achieved impressive feats as generative paradigms for continuous data, such as images and videos, and more recently, for high-dimensional discrete data. Despite this, multimodal generation combining discrete and continuous modalities remains dominated by autoregressive models that work on discrete tokenized inputs or continuous projected embeddings. In this work, we present CrossFlows, a new paradigm for multimodal generation that learns a flow on a joint discrete and continuous space. We show CrossFlows are capable models for multimodal generation, as well as text-to-image, image-to-text generation and other single-modality and multimodal downstream tasks.

## 1 INTRODUCTION

Multimodal generation and understanding, defined as the processing of both continuous and discrete inputs and outputs, is the next frontier of Generative AI Zhang et al. (2025); Wang et al. (2023). Applications range from interleaved multimedia generation to scientific use cases such as protein design Campbell et al. (2024), or robotics Black et al. (2024).

Classical and current state-of-the-art approaches include combining various pre-trained encoders with a language decoder Ye et al. (2023); Gemini (2025); Wang et al. (2024c); Wu et al. (2024); Bai et al. (2025); Chen et al. (2025), as well as pre-training a single decoder on both image and text tokens, an approach known as early-fusion training Zhu et al. (2025); Llama-4 (2025); Chameleon (2025); Wang et al. (2024d).

All these solutions rely on a powerful autoregressive discrete decoder. Despite impressive results, these architectures are limited by sampling inefficiency Jayaram & Thickstun (2021) and modality asymmetry Parcalabescu & Frank (2024) which, for instance, leads to visual information being disregarded in favor of textual biases Vo et al. (2025). Furthermore, it has been shown that vision-language models become less reliant on visual information if sufficient textual clues are provided Wang et al. (2024b), and that they struggle with tasks that require pure visual understanding Fu et al. (2025); Ramakrishnan et al. (2025).

These shortcomings can be partly attributed to the image tokenization approach common in many of these architectures, as evidence suggests that while discrete semantic tokens excel at capturing high-level patterns, they sacrifice fine-grained spatial information Baek et al. (2025); Liu et al. (2024); Yan et al. (2024).

There are some alternatives to the autoregressive paradigm for generation. Flow Matching (FM) models learn a vector field that transports a simple source distribution to a target data distribution Lipman et al. (2022). On the other hand, Diffusion models learn to reverse a diffusion process that gradually adds noise to real data, effectively learning to parameterize a reverse-time Stochastic Differential Equation that maps from the source distribution to the target distribution Ho et al. (2020). While on continuous modalities Flow and Diffusion models frequently outperform autoregressive transformers Prajwal et al. (2024); Cai et al. (2025), and match their performance on discrete modalities Nie et al. (2025); Gat et al. (2024) (excelling in abstract rule learning Wang et al. (2024a)), few attempts have been made to use them for multimodal generation.

Both Yang et al. (2025) and Swerdlow et al. (2025) train a Large Diffusion Model on discrete image and text tokens, showing strong performance on textual-reasoning and text-to-image generation, but

failing to leverage the continuous nature of image-data. Xie et al. (2024) combine Discrete Diffusion with Autoregressive generation into a single transformer, training autoregressively on text tokens and on masked tokens on the image part. Perhaps most notably, Campbell et al. (2024) train a joint discrete and continuous Flow model for protein structure generation, achieving *true* multimodal generation on both continuous and discrete spaces and laying the foundation for a discrete flow-based model based on Continuous Time Markov Chains (CTMCs).

These advancements gain further significance when analyzed under the light of the Generator Matching framework Holderrieth et al. (2025), which provides the theoretical foundations for multimodality in generative models. In particular, this work shows how flow and diffusion models are particular cases of the more general problem of parametrizing the generator of a Markov process, and goes on to prove that generators can be linearly combined, which allows for multimodal generation, as well as the combination of flow, diffusion, and other stochastic generative models such as jump models.

In this work, we build on the Generator Matching framework to train an image-language multimodal generative model, showing that flow models can provide an effective alternative to the autoregressive paradigm for multimodal generation. Instead of mapping both modalities to a common space, we learn a factorized probability path on the continuous space of images and the discrete space of language tokens, which allows us to achieve coupled cross-modal generation. That is, the generation of each modality is influenced by the evolving state of the other modality, not just conditioned on a fixed input. Further, we show this approach is equivalent to implicit guidance, which allows us to control generation, while at the same time mitigating some of the limitations of current approaches such as Classifier Free Guidance. Our contributions can be summarized as follows:

- We introduce the CrossFlows architecture for joint discrete and continuous multimodal generation.
- We define the multi-modal theory foundation for our framework under the Generator Matching theory.
- We show CrossFlows allow for implicit (manifold-constrained) guidance, making them flexible foundational models for multimodal generation, single modality and other downstream tasks such as image inpainting.

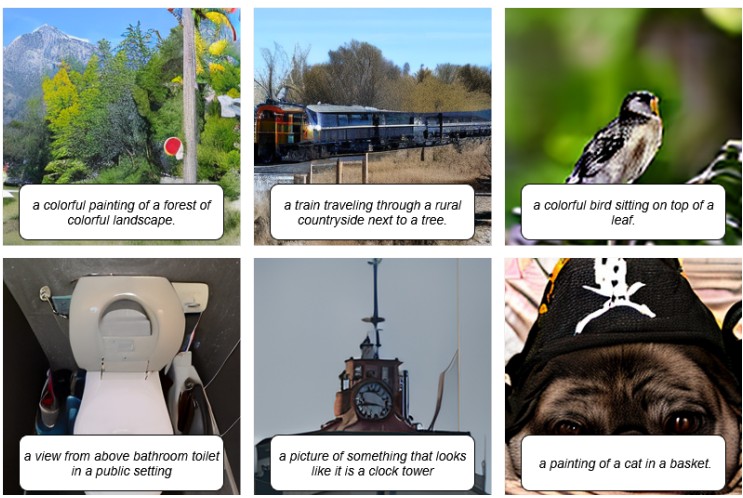

Figure 1: Unconditional generation of Image/Caption pairs with CrossFlows.

## 2 BACKGROUND

### 2.1 CONTINUOUS FLOWS

Let $\psi_t : \mathbb{R}^d \to \mathbb{R}^d$, $\psi_t \in C^r(\mathbb{R}^d, \mathbb{R}^d)$ be a $C^r$ dipheomorphism, i.e.: a continuously ($r$ times) differentiable and invertible map such that $\psi_t^{-1} \in C^r(\mathbb{R}^d, \mathbb{R}^d)$.

Now let $\psi : [0,1] \times \mathbb{R}^d \to \mathbb{R}^d$ such that $\psi(t,x) \mapsto \psi_t(x)$. The function $\psi$ is called a $C^r$ *flow*. Given a random variable (r.v.) $X_0 \sim p_0$, a *flow model* is a continuous time Markov process $(X_t)_{0 \le t \le 1}$ such that

$$X_t = \psi_t(X_0), \; X_0 \sim p_0.$$

We can also express the above Markov process via an ODE:

$$\frac{d}{dt}\psi_t(x) = u_t(\psi_t(x)), \;\; \psi_0(x) = x,$$

where $u$ is a $C^r([0,1] \times \mathbb{R}^d, \mathbb{R}^d)$ velocity field $u : [0,1] \times \mathbb{R}^d \to \mathbb{R}^d$, $u(t,x) \mapsto u_t(x)$. Existence and uniqueness of solutions to this ODE can be shown up to time $t = 1$ (almost everywhere) as long as we assume global Lipschitzness or integrability of $u$ (see Lipman et al. (2024)).

Such a flow defines a time dependent probability path $(p_t)_{0 \le t \le 1}$, $X_t = \psi_t(X_0) \sim p_t$ for $X_0 \sim p_0$. Conversely, given a probability path $p_t$, we say $u_t$ generates $p_t$ if $X_t = \psi_t(X_0) \sim p_t$. Notice $(p_t)_{0 \le t \le 1}$ is a Markov process, because if we consider a partition $0 \le t_1 < \ldots < t_n < 1$, then for a Borel set $A$:

$$(X_{t_n+dt} \in A \mid X_{t_1}, \ldots, X_{t_n}) = (X_{t+dt} \in A \mid X_{t_n}).$$

Under this setting, training a *flow matching model* amounts to

1. building a probability path $(p_t)_{0 \le t \le 1}$ generated by a flow $u_t$ such that $p_1$ aligns with our target distribution and,

2. finding a parametrization $u_t^\theta$ of $u_t$ such that it minimizes the loss

$$L_{FM}(\theta) = \mathbb{E}_{t \sim U(0,1), X_t \sim p_t}\left[\|u_t(X_t) - u_t^\theta(X_t)\|^2\right],$$

where instead of the L2-norm we could use any Bregman divergence, see Lipman et al. (2024); Holderrieth et al. (2025). The issue here is that $u_t$ is not tractable. Luckily, it suffices to optimize with respect to the conditional flow. Given a r.v. $Z \sim p_1$, the conditional loss is defined as,

$$L_{CFM}(\theta) = \mathbb{E}_{t, Z, X_t \sim p_{t|Z}(\cdot|Z)}\left[\|u_t(X_t \mid Z) - u_t^\theta(X_t)\|^2\right],$$

where $u_t(x \mid Z = z)$ is a conditional vector field generating the conditional probability path $p_{t|1}(\cdot \mid z)$, from which it is possible to extract the marginal probability path:

$$p_t(x) = \int p_{t|1}(x \mid z)p_1(z)dz.$$

It can be proven that under mild assumptions, and if $u_t(x \mid z)$ is conditionally integrable and generates $p_{t|1}(\cdot \mid z)$, then the marginal velocity field $u_t$ generates the marginal probability path $p_t$. Further, optimizing $\theta$ for $\mathcal{L}_{CFM}$ is equivalent to optimizing for $\mathcal{L}_{FM}$, given that the gradients of both losses coincide (refer to Lipman et al. (2024); Holderrieth et al. (2025) for proofs of both facts).

## 2.2 DISCRETE FLOWS

Suppose now we aim at building a flow model to generate discrete samples $x \in \{1, ..., S\}^{\tilde{d}}$. Consider a rate matrix $R_t \in \mathbb{R}^{S \times S}$ such that the probability of jumping from state $x_t$ to state $j$ at time $t$ is given by

$$p_{t+dt|t}(j|x_t) = \delta_{x_t}(j) + R_t(x_t, j)dt$$

where $\delta_{x_t}$ is the Kronecker delta (1 if $j = x_t$ and 0 otherwise) and $R_t(j,j) = -\sum_{j \ne k} R_t(j,k)$.

A trajectory can then be simulated taking Euler steps $x_{t+h} \sim \delta_{x_t}(\cdot) + R_t(x_t, \cdot)ht$, starting with $x_0 \sim p_0$. In practice, we parametrize the rate matrix $R_t$ and perform updates component-wise (see Gat et al. (2024))

$$X_{t+h}^i \sim \delta_{X_t^i}(\cdot) + hu_t^{\theta i}(X_t, \cdot).$$

As in the continuous case, rate matrix $R_t$ generates the probability path $p_{t|1}(\cdot \mid z)$ if and only if it satisfies the *Kolmogorov Forward Equation*:

$$\partial_t p_t(x_t) = \sum_{j \neq x_t} R_t(j, x_t) p_t(j) - \sum_{j \neq x_t} R_t(x_t, j) p_t(x_t).$$

Discrete Flows where first introduced in Campbell et al. (2024), including the uniform sampling probability path for token interpolation,

$$p_{t|1}(x_t \mid x_1) = f(t)\delta_{x1}(x_t) + (1 - f(t))\frac{1}{S},$$

where $f(t)$ is such that $f(0) = 0$, $f(1) = 1$ and monotonously increasing, and $p_0$ is the uniform distribution over tokens. We simulate this probability path in our implementation of discrete flow matching, see Section 3.3.

## 2.3 GENERATOR MATCHING FOR MULTIMODAL GENERATION

The Generator Matching framework introduced by Holderrieth et al. (2025) provides a unified language for understanding generative modeling based on Markov processes, including *flows*, *diffusion models* and *jump models*, while providing powerful theoretical insights that allow the combination and superposition of generators.

Generally speaking, given a Markov process $(X_t)_{0 \leq t \leq 1}$ with transition kernel $(k_{t+h|t})_{0 \leq t < t+h \leq 1}$ (i.e.: $[X_{t+h} \in A \mid X_t = x] = k_{t+h|t}(A \mid x))$, and for any test function $f \in C_c^\infty$, then its generator $\mathcal{L}_t$ is defined as

$$[\mathcal{L}_t f](x) = \frac{d}{dh}\bigg|_{h=0} \int f(z) k_{t+h|t}(dz \mid x)$$

$$= \frac{d}{dh}\bigg|_{h=0} \mathbb{E}_{z \sim k_{t+h|h}(\cdot|x)}[f(X_{t+h}) \mid X_t = x],$$

which acts as a sort of derivative of the transition kernel. Holderrieth et al. (2025) goes on to show that a generator $\mathcal{L}_t$ of a Markov process satisfies the *Kolmogorov Forward Equation*

$$\frac{\partial}{\partial t}\mathbb{E}_{x \sim p_t}[f(x)] = \mathbb{E}_{x \sim p_t}[\mathcal{L}_t f(x)], \tag{KFE}$$

if and only if $X_t$ generates the probability path $(p_t)_{0 \leq t \leq 1}$. This generalizes generative modeling as parametrizing a generator $\mathcal{L}_t$. For example, a *flow* as described in the previous subsection has generator $\nabla f u_t$. As explained for flows, the marginal object is not tractable, but conditional generators $\mathcal{L}_t^z$ (generating conditional probability paths $p_t(\cdot \mid z)$) are, which enables scalable training.

The fact that the KFE is a linear equation and the generator a linear operator has deep reaching implications for multimodal generation. Let $q_t^1(\cdot \mid z_1)$, $q_t^2(\cdot \mid z_2)$ be two conditional probability paths on $S_1$, $S_2$ generated by processes $I_t$ and $C_t$ with generators $\mathcal{L}_t^{\tilde{z}_1}$ and $\mathcal{L}_t^{\bar{z}_2}$ resp. Define a conditional factorized path on $S_1 \times S_2$ as $p_t(\cdot \mid z_1, z_2) = q_t^1(\cdot \mid z_1)q_t^2(\cdot \mid z_2)$. Then the Markov process $X_t = (I_t, C_t)$ with generator

$$\mathcal{L}_t^z f(x_1, x_2) = [\mathcal{L}_t^{\tilde{z}_1} f^{x_1}](x_1) + [\mathcal{L}_t^{\bar{z}_2} f^{x_2}](x_2),$$

(where $f$ is a test function on $S_1 \times S_2$ and $f^{x_i}$ its restriction to $S_i$) is a solution to the KFE, i.e: it generates $p_t(\cdot \mid z_1, z_2)$. See Holderrieth et al. (2025) for a detailed proof.

This means that the linear combination of the generators of a discrete process $C_t$ and a continuous process $I_t$ generates the probability path $p_t$ on the joint multimodal space. We will leverage this fact in the next section to learn probability paths in the product of discrete and continuous spaces.

# 3 MULTIMODAL GENERATION (OUR APPROACH)

## 3.1 MULTIMODAL FLOWS

Let $S = S_1 \times S_2 = \mathbb{R}^d \times \{1, ..., T\}^{\tilde{d}}$ be a multimodal space that combines continuous and discrete data. Let $p_1$ be the underlying distribution, approximated by an image-text dataset $\mathcal{D}$. Let $z =$

$(i_1, c_1) \sim p_1$ be a pair image-caption. Define factorized probability paths as

$$p_{t|1}(x_t \mid z) = q^1_{t|1}(x_t \mid i_1)q^2_{t|1}(x_t \mid c_1),$$

and let $u_t(\cdot \mid z)$ be a flow generating $p_{t|1}(\cdot \mid z)$ as above. In our case, $q_1$ is the CondOT path, and $q_2$ the uniform sampling probability path described in Section 2.2.

In order to learn a parametrization of $p_t$ via two model components, $u^{\theta_1}_t$ continuous and $u^{\theta_2}_t$ discrete, we make each component depend on the multimodal state $Z_t = (I_t, C_t)$, so that we approximate the flow $u_t(\cdot \mid z)$ generating $p_{t|1}(\cdot \mid z)$ by $u^\theta_t = (u^{\theta_1}_t(\cdot \mid Z_t), u^{\theta_2}_t(\cdot \mid Z_t))$.

This results on an interdependent generation process across modalities, meaning that both modalities are generated in a coordinated way and their outputs are mutually consistent and sampled from the joint distribution: $(I_t, C_t) \sim p_t(\cdot \mid I_0 = x, C_0 = y) \implies (x, y) = \left(\psi^{-1(1)}_t(I_t), \psi^{-1(2)}_t(C_t)\right)$. This guarantees a stronger consistency between modalities than autoregressive models, which often generate one modality first and then the other one conditionally. In other words, since flows are invertible and we are sampling from a true joint latent, we are guaranteeing consistency between modalities.

This allows us to define the *Multimodal Flow Matching Loss* as

$$L_{MM}(\theta) = \mathbb{E}_{t,Z_1,I_t \sim p_t(\cdot|I_1),C_t \sim p_t(\cdot|C_1)} \left[D_1 + D_2\right],$$

where

$$D_1 := \|u^1_t(I_t \mid Z_1) - u^{\theta_1}_t(I_t, C_t)\|^2, \quad D_2 := \|u^2_t(C_t \mid Z_1) - u^{\theta_2}_t(I_t, C_t)\|^2.$$

As explained by Holderrieth et al. (2025), optimizing the linear combination of losses $D_1$ and $D_2$ is equivalent to learning the conditional joint generator and, by marginalization, the generator of the multimodal process $X_t \sim p_t$ such that $X_1 = (I_1, C_1) \sim p_1$ is a pair image-caption.

## 3.2 Guidance and Cross-Guidance

Let $y$ be a guiding signal from a space $S_2$ which we want to use to guide generation (such as a text prompt) for a flow model $u_t$ generating probability paths $p_t$ in a space $S_1$. As long as we can sample from the conditional target distribution $p_{1|Y}(\cdot \mid y)$, we can construct guided probability paths as

$$p_{t|Y}(x \mid y) = \int p_{t,1|Y}(x, z|y)dz = \int p_{t|1}(x|z)p_{1|Y}(z|y)dz$$

where $p_{t|1}(x \mid z)$ is the conditional path that we saw on the previous section, which does not depend on $y$. Similarly, the guided velocity field is given by

$$u_{t|Y}(x \mid y) = \int u_t(x \mid z)\frac{p_{t|1}(x \mid z)p_{1|Y}(z \mid y)}{p_{t|Y}(x \mid y)}dz.$$

Using Classifier Free Guidance Ho & Salimans (2022), we can construct the parametrization simply as

$$\tilde{u}^\theta_t(x \mid y) = (1 - \omega)u^\theta_t(x \mid \emptyset) + \omega u^\theta_t(x \mid y).$$

Despite the fact that $\tilde{u}^\theta_t$ does not exactly generate the path $(p_t)_{0 \le t \le 1}$, it remains the most widely applied guidance method.

Generally, the conditional probability path $p_{t|Y}(\cdot \mid y)$ is defined using pairs image-captions $\in S_1 \times S_2$. During sampling, the guiding signal $y \in S_2$ is kept unperturbed, while $x_t \sim p_{t|Y}(\cdot \mid y)$. This allows for controlled generation, but raises the question: how close are $p_1(x) = \mathbb{E}_{y \sim Y}[p_{1|Y}(x \mid y)]$ and $\mathbb{E}_{\cdot,y \sim \mathcal{D}}[p_{1|Y}(x \mid y)]$? That is, can we generate most samples in $p_1$ with the appropriate prompt? Or does Classifier Free Guidance hamper interpolation?

There is some research in this direction. The phenomenon of mode collapse, that is, diversity loss from over-emphasis on high-likelihood regions is well known in the literature, and specially acute for higher values of $\omega$. In particular, Chung et al. (2024) tie mode collapse to the guiding signal forcing generation out of the data manifold. It has also been observed by Patel et al. (2023).

As we will show, multimodality can help mitigate this issue.

**Cross-Guidance.** In the multimodal setting, we construct the guided velocity field as $u_{t|I_t,C_t}(x \mid i_t, c_t)$ which generates the probability path $p_{t|1}(x_t \mid z)$, as explained in Section 3.1. Under this approach, guidance happens implicitly: instead of conditioning on a fixed input, the generation of each modality is influenced by the evolving state of the other modality.

This allows for greater flexibility. Given a signal $y$ as before (e.g.: a text prompt), then the guided probability path is

$$p_{t|Y}(x \mid y) = \int p_{t|1}(x \mid z)p_{1|Y}(i_1 \mid c_1 = y)dz,$$

because $p_{1|Y}(z|y) = p_{1|Y}(i_1 \mid c_1 = y)$ in the multimodal setting. But we don't want to generate $p_{t|Y}$. Instead, we will generate $p_{t|C_0}(\cdot \mid c_0 = y)$. That is, instead of starting from $C_0 \sim q_0^2 = Unif([vocab\_size]^{\tilde{d}})$ we start with $c_0 = y$. Then, along the multimodal probability path $p_t$, this signal will influence $I_t$, while being allowed to be changed towards the data distribution (if needed). Effectively, this means Cross-Guidance acts as Manifold Constrained guidance Chung et al. (2024). Moreover, as the evolution of $C_t$ is stochastic (for it depends on $I_0 \sim q_0^1$, usually a Gaussian distribution) it helps mitigate mode collapse. More details on Cross-Guidance and experiments on the Appendix C.

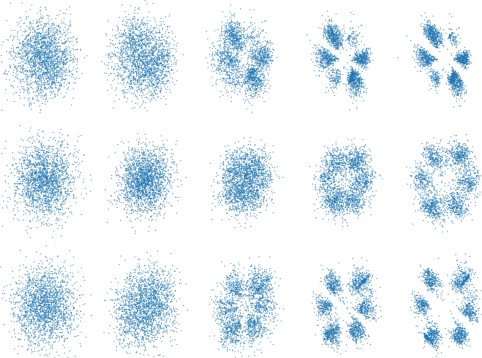

Figure 2: Experiment illustrating the phenomenon of mode collapse. The target distribution corresponds to six gaussians, each point assigned a label that corresponds to its position on a grid, i.e.: x and y indices of the box that contains it added together, which implies labels along the top-right, bottom-left diagonal are underrepresented. When sampling using all labels, CFG (first row), misrepresents those regions. CrossFlows (second row) doesn't, but leads to more noisy predictions. Third row represents CrossFlows + CFG.

### 3.3 ARCHITECTURE

Our architecture consists of two jointly trained sub-models, one per modality. The text-to-image (continuous flow) model is based on DiT Peebles & Xie (2023), originally designed for class conditioning but extended here to handle both text and image inputs. It uses MM-DiT blocks along with patch, timestep, and context embeddings, plus final linear layers. The image-to-text (discrete flow) model is a transformer with cross-attention layers that fuse image embeddings with text.

**Continuous Flow Model.** Let $\mathcal{D}$ be a dataset of image–caption pairs $(I, C)$. The text-to-image model operates in the latent space, where images $I \in \mathbb{R}^{B \times H \times W \times C}$ are encoded by a VAE into latent representations $Z \in \mathbb{R}^{B \times H/8 \times W/8 \times 4}$. These latents are first processed by a learnable patch embedding layer, which partitions the input into non-overlapping patches of size *patch size* × *patch size* and projects each patch into a $D$-dimensional vector. This produces a sequence of embedded patches:

$$\mathbf{x}_{\text{embed}}^i \in \mathbb{R}^{B \times T \times D}, \quad T = \frac{(H/8)(W/8)}{\text{patch size}^2}, \quad D = \text{hidden size}$$

Additional embeddings are introduced before forwarding the sequence to the backbone. The diffusion timestep $t$ is mapped to a $D$-dimensional vector via an MLP, while conditioning signals (e.g., text context) can be projected to the same dimension when required. A fixed 2D sinusoidal positional encoding is also added to the input sequence.

Captions $C$ are processed as conditioning information. They are tokenized with the CLIP tokenizer and encoded using a pre-trained CLIP text encoder. The backbone follows the MM-DiT architecture, which fuses image patches, text embeddings, and timestep embeddings through cross-attention. The number of MM-DiT blocks is determined by a depth parameter. Finally, a lightweight MLP projects the features before the *unpatchify* operation reconstructs a tensor of the same size as the latent input.

**Discrete Flow Model**. Using the same dataset as in the continuous model, $\mathcal{D}$ with pairs $(i, c)$ of images and captions of length $T$, where $i \in \mathbb{R}^{B \times H \times W \times C}$ and $c \in \mathbb{N}^{B \times T}$, we now treat captions as the training data and images as the conditioning factor for a discrete flow matching model. The images are encoded using a pretrained image encoder (dinov2vitb14, Oquab et al. (2023)), resulting in a sequence of image tokens $z \in \mathbb{R}^{B \times S \times D}$, where $S$ is the number of visual tokens and $D$ is the embedding dimension. Then, each caption token $c$ is embedded into a dense vector space using a learned embedding layer, producing a sequence of text token embeddings $\mathbf{x}^c_{\text{embed}} \in \mathbb{R}^{B \times T \times D}$, where $T$ is the number of tokens in each caption and $D$ is the hidden size.

Similar to the continuous case, a timestep embedding gets the time value $t$ and generates a conditioning vector, using a frequency-based sinusoidal embedding and an MLP projection, modulating attention and MLP layers using an adaptive layer norm (AdaLN). $\mathbf{x}_{\text{embed}}$ are then passed through a transformer attending to the image tokens $z$ and using AdaLN conditioning with the timestep embedding $c_t = \text{MLP}(f_t) \in \mathbb{R}^{B \times D_c}$ for $D_c$ the image conditioning dimension. This process is repeated for a number of blocks, where each one refines the input via:

$$x \leftarrow \text{CrossAttn}(x, z) + \text{SelfAttn}(x) + \text{MLP}(x),$$

after which the final layer of the model projects the refined token embeddings back to the vocabulary logits.

## 3.4 TRAINING

A schematic overview of the training pipeline is shown in Figure 3. We begin with a batch of dataset instances consisting of images $I_1$ and captions $C_1$, where $Z_1 = (I_1, C_1) \sim p_1$ follows the real data distribution. The images are encoded into the latent space using a VAE, and additional image features are extracted with the encoder. We denote the latent representation as $I^z$ and the image embedding as $I^e$. Next, a batch of timesteps is sampled, and the probability path is applied to obtain $Z_t = (I^z_t, C_t)$, where both domains have the same level of noise.

The continuous flow model takes as input the latent representation $I^z_t$, the text embedding of the prompt $C_t$, and the timestep. To improve training efficiency, we employ several optimization techniques. Using $I^e$, we apply REPA alignment Yu et al. (2025) to compare the original image embeddings with the features extracted from the 8th MM-DiT block. Additionally, we incorporate HASTE Wang et al. (2025), which aligns both image embeddings and attention maps. In addition, Classifier-Free Guidance is applied within the image model. For the image domain, we adopt the classical conditional optimal transport path as the probability path.

The discrete flow model receives the text embedding of the prompt $C_t$, the image embedding of the image $I^e_t$ and the timestep. The model is inspired by Markov jump processes, where the learned transition function is factorized across the token dimensions. Each dimension $d \in [1, \ldots, T]$ is stochastically assigned a jump schedule, $t^*_d$ (i.e. a time when it is allowed to change), and at each timestep $t$, the model outputs logits for every dimension, updating the dimension with the earliest scheduled time $t^*_d \geq t$ and leaving the rest unchanged. Once this dimension has been updated, a new schedule is updated and this process continues until the scheduled time is 1 for all dimensions, i.e. $t^*_d = 1 \forall d \in [1, \ldots T]$. For the text domain, we employ the MixtureDiscreteProbPath with a polynomial convex scheduler. The source distribution is defined either using mask tokens or a uniform distribution over the vocabulary. Training of the text model is guided by two different options of objectives: cross-entropy loss and the GeneralizedKL loss.

During training, we evaluate multiple multimodal generation configurations, with particular emphasis on the effect of incorporating pre-trained components (a pre-trained T2I model and/or a pre-

trained I2T model). Finally, for optimization under the joint loss defined in Equation 3.1, we apply a weighted combination of the image and text loss terms.

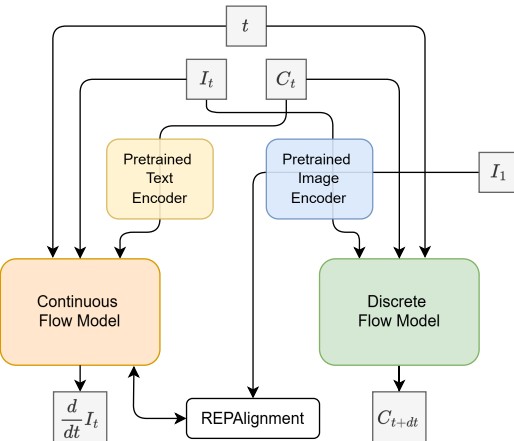

Figure 3: CrossFlows architecture and training strategy. We sample a time $t$, and interpolate on discrete and continuous probability paths to generate image and text interpolations $I_t$ and $C_t$, which are then passed as input to both the continuous and discrete flow models. We can also perform REPA Alignment on the images to improve the quality of generations.

## 3.5 Sampling

Sampling is performed parallelly on the discrete and continuous flows. We begin by sampling $Z_0 = (I_0, C_0) \sim p_0$ and a time grid, and proceed to solve both ODE's via an Euler method, feeding in each step $I_t$ and $C_t$ to both models. Guided generation is performed starting with $\tilde{Z}_0 = (I_0, \text{prompt})$ or $\bar{Z}_0 = (\text{image}, C_0)$. More details as well as other sampling strategies on Appendix A.1.

## 4 Experiments & Results

### 4.1 Experimental setup

We evaluate the effectiveness of our method through a series of experiments. All models are initially trained on the COCO2017 dataset , which contains approximately 100k images, each paired with five captions. In configurations that incorporate pre-trained components, we additionally use the CC3M dataset, which provides about 2.9M image–caption pairs for training the T2I and I2T models.

For evaluation, we generate 4k synthetic samples from the trained models and compare them against an equal number of samples drawn from the COCO2014 validation set, which contains roughly 50k samples in total. We assess performance across three tasks: image generation (FID, FLD, Precision, Recall), text generation / image captioning (Perplexity, Cosine Similarity), and multimodal generation (CLIPScore). We add Image Inpainting experiments in Appendix C.2.

### 4.2 Results

**Image generation**. We evaluate the quality of images generated by the continuous flow model trained in the multimodal setup. The results are reported in Table 4.2, where we compare our proposed multimodal model against a flow-matching baseline trained with HASTE, the CC3M dataset, and an MM-DiT backbone. When trained from scratch, the multimodal model shows a reduction in image quality, as indicated by a higher FID score. To improve performance, we initialized the image model with a pre-trained T2I model; however, this configuration performed worse than both the baseline and the scratch-trained multimodal model.

**Text generation (image captioning)**. Analogously to the text-to-image evaluation, we evaluate the text generation capabilities by producing captions conditioned on 4k real images selected ran-

| Method | FID ↓ | Precision ↑ | Recall ↑ | ClipScore ↑ | Caption Sim ↑ |
|---|---|---|---|---|---|
| FM-MMDiT-HASTE (T2I Model) | **22.12** | - | - | 17.21 | - |
| Cross-Modal FM + REPA | 25.48 | 0.68 | 0.53 | **20.30** | 0.2694 |
| Cross-Modal FM + Pre-trained T2I | 27.38 | 0.68 | 0.52 | 19.81 | **0.6185** |

Table 1: Quantitative results for different models. For image quality, lower FID is better and higher Precision and Recall are better. For image-text alignment we have the ClipScore where higher value is better.

domly from the COCO2014 validation set. Then, we compare the similarity between real and synthetic captions using the cosine similarity of their sentence embeddings, obtained with the sentence-transformer model UKP Lab. The resulting mean cosine similarity is 0.61 (with a standard deviation of 0.16), indicating a moderate-to-strong overall alignment with the real captions.

**Multi-modal generation**. After assessing the capabilities in image and text generation separately, it's time to study the conceptual alignment of text and images when generated together. The Clip-Score values are presented in Table 4.2 where we can see an improvement of the alignment compared to the baseline using the multi-modal setup. Figure 4 compares the distribution of ClipScores between samples from the validation set of COCO2014 and a synthetic dataset generated with Cross-Flows, suggesting a semantic match between image and captions similar to real datasets.

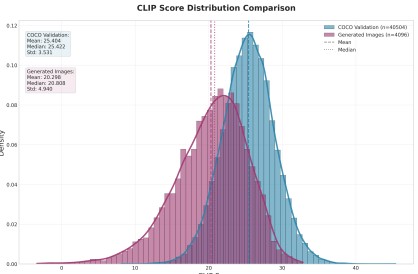 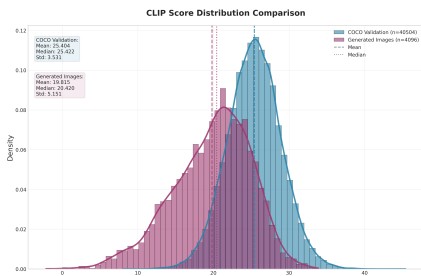

Figure 4: Comparison of CLIPScores between 50K image-caption pairs generated by CrossFlows and those from the ground-truth validation set of COCO2014, alongside another distribution comparison.

## 5 LIMITATIONS & FUTURE WORK

The experiments in this work have been carried out with a small scale dataset (around $110k$ images and $7.5k$ text tokens). It is not the scope of this paper to compare against state-of-the-art generative models, for which large scale pre-training with millions of images and billions of tokens would be needed, but to showcase the potential of flow models as an alternative to autoregressive multimodal generation. Future work includes larger scale training as well as interleaved image-text-other modalities generation.

## 6 CONCLUSIONS

In this paper, we introduced CrossFlows, a new paradigm for multimodal generation that learns a flow on a joint discrete and continuous space. Rather than mapping different modalities into the same space, we learn probability paths on their product space. We name this approach Cross-Modal generation, as the evolving state of each modality depends on the distribution of the other modality. This entails some advantages: it avoids modality asymmetry, as we don't need a continuous or discrete decoder; it permits conditional generation on unfixed inputs, which mitigates mode collapse and lack of diversity of generations; and it permits solving downstream tasks previously hard to solve with Flow models, such as image inpainting. We hope this work will contribute to the progress toward new paradigms for multimodal generation.

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

# APPENDIX

## A  TRAINING & EVALUATION DETAILS

### A.1  SAMPLING

We employ the following strategies to generate high-fidelity synthetic data pairs of images and captions.

**Naive Sampling**. The easiest way to sample is to directly follow the same logic as in training, that is, doing generation sequentially following the inference method of each model, and conditioning the next update on the current image/caption interpolation prediction. Thus, one can start by sampling from the source distributions of both modalities (random Gaussian noise for the images and uniform tokens for the text), obtaining $(I_0, C_0) \sim p_0$, as well as defining a discretization of the time interval $t \in [0, 1]$ with a set number of *steps*. See Algorithm 1.

---

**Algorithm 1** Multimodal generation sampling

---

**Input**: $Z_0 = (I_0, C_0) \sim p_0$
**Parameter**: $u_1(I_t, C_t, t)$, $u_2(C_t, I_t, t)$, $n$, $w_{cfg}$
**Output**: $Z_1 = (I_1, C_1) \sim p_1$

    Let $t = 0$.
    Let $h = 1/n$
    **while** $t \leq 1$ **do**
        **if** $w_{cfg} \geq 1$ **then**
            $v_1 = u_1(I_t, C_t, t)$, $v_2 = u_2(C_t, I_t, t)$
            $v_{1,\emptyset} = u_1(I_t, \emptyset, t)$
            $v_1 = v_{1,\emptyset} + w_{cfg} \cdot (v1 - v_{1,\emptyset})$
        **else**
            $v_1 = u_1(I_t, C_t, t)$, $v_2 = u_2(C_t, I_t, t)$
        $I_{t+h} = I_t + h \cdot v_1$
        $C_{t+h} = h \cdot v_2$
        $Z_{t+h} = (I_{t+h}, C_{t+h})$
        Let $t = t + h$
**return** $Z_1 = (I_1, C_1)$

---

**Adaptive Horizon Sampling**. A more interesting way of sampling is to leverage the number of steps it takes for one modality to reach stability, and allow the generative process each modality depend on this factor. Essentially, we introduce a horizon parameter for the evolution of each modality, allowing inference over a time interval rather than only allowing updates on pairs of steps $(t, t_1)$. As we noticed in our experiments that the text sampling stabilized in a fewer number of steps than the images, we decided to set a bigger sampling horizon for the image generation, allowing them to evolve through more timesteps before conditioning the next text update.

We observe Adaptive Horizon Sampling to be beneficial, as bad text quality during the initial steps hinders image definition. Adaptive Horizon Sampling allows the text to gain semantic significance faster. However, we have not yet fully investigated whether this leads to more pronounced mode collapse.

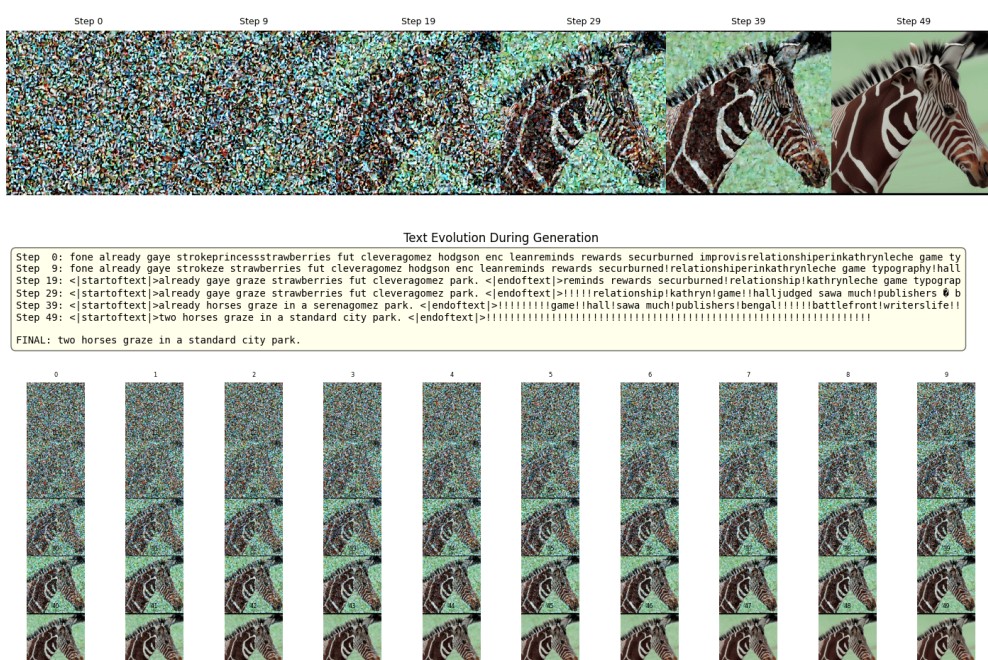

Figure 5: Unconditional sampling of an Image/Caption pair with CrossFlows.

## A.2 TRAINING DETAILS

We train our models on a single node with data-parallel distributed training across four NVIDIA H100 GPUs using the Accelerate framework, for a total of 250-400k steps. Training is performed with a batch size of 256 using the AdamW optimizer, with a learning rate of $1 \times 10^{-4}$ and a weight decay of $1 \times 10^{-5}$.

## A.3 PERPLEXITY

One of the advantages of Flow Matching models is that they allow for explicit estimation of log probabilities Lipman et al. (2022). This is specially interesting in the case of discrete flow matching, as it allows to estimate the quality of text generation, without resourcing to n-gram based metrics for text comparison such as BLEU or ROUGE, which do not capture the semantic meaning of the text.

Let $I_1$ be a sample image, and $C_{pred}$ a generated caption. Following Haxholli et al. (2024), we may write

$$- \log_{p_1}(C_{pred}; \theta) \leq \int_0^1 \frac{\kappa_t'}{1 - \kappa_t} \sum_{x_t} p_{t|1}(x_t \mid x_1)$$

$$\sum_{i=1}^L (\mathbb{1}_{\{x_1^i \neq x_t^i\}} \log p_{1|t}^i(x_1^i \mid x_t; \theta) + 1 - p_{1|t}^i(x_t^i \mid x_t; \theta) - \mathbb{1}_{\{x_1^i \neq x_t^i\}}) dt$$

The integral may be approximated via Monte Carlo methods, but the difficulty strives in the sum over $x_t$: depending on the text schedule chosen, the number of possibilities can be too great. There are two ways around this: choose a schedule in which only one token changes by timestep, or estimate $p_{t|1}(x_t \mid x_1)$ by computing only backward probability paths from $x_1$, and adding some noise, as these are the instances of $x_t$ with highest probability, and the others in the sum can be bounded with these. We follow the second approach.

# B DIFFUSION

Diffusion models Ho et al. (2020) are generative models trained to learn the reverse of a diffusion process that gradually adds noise to real data. At generation time, they start from pure noise and progressively denoise it to produce a realistic sample. We investigate here the potential of diffusion models as backbone for our multimodal architecture.

The fundamental concept and the key distinction from other latent variable models involves a standard forward corruption process $q$, which takes clean data $x$ from the data distribution $q(x)$ and defines a latent variable $z_t$ for $t \in [0, 1]$, representing progressively noisy versions of $x$ as follows:

$$z_t = \sqrt{\alpha_t} x + \sqrt{1 - \alpha_t} \epsilon$$

where $\epsilon \sim \mathcal{N}(0, \mathbf{I})$ and $(\alpha_t)_{t \in [0,1]}$ is a noise schedule, monotonically decreasing in $t$ Sahoo et al. (2024). Therefore, it is a predetermined Markov chain that incrementally adds Gaussian noise to the data.

On the other hand, the reverse diffusion model $p_\theta$ parameterized over $x$ and $z_t$ is trained to maximize a variational lower bound on log-likelihood (ELBO), conceptualizing it as a Markov chain with Gaussian transitions.

## B.1 DISCRETE DIFFUSION

Discrete Diffusion Models can be broadly categorized into two types based on whether they embed discrete structures in a continuous space and then perform a Gaussian diffusion Chen et al. (2023); Li et al. (2022), or if they define a diffusion process directly on discrete structures. D3PM Austin et al. (2021) presents a framework that includes a Markov forward process that is characterized by the multiplication of matrices $Q_t$ over T discrete time steps as described below Sahoo et al. (2024):

$$q(z_t|x) = Cat(z_t; \overline{Q}_t x) = Cat(z_t; Q_t \cdot Q_{t-1} \cdots Q_1 x)$$

Thus, the scoring viewpoint on diffusion modeling may be extended to discrete data using the Continuous Time Markov Chain (CTMC).

## B.2 Simple Masked Diffusion Models

The authors in Sahoo et al. (2024) note that, rather than supporting general noise processes, like those used in discrete diffusion models, absorbing state diffusion (where the unmasked token remains unchanged during the reverse diffusion and the clean input is never masked) delivers superior performance. Therefore, they focus on masking techniques and develop a precise Rao-Blackwellized approach that does not rely on CTMC theory, achieving better results than general methods in this area.

## B.3 Masked Diffusion Language Models (MDLM

Masked Diffusion Language Models (MDLM) represent a significant advancement in discrete diffusion modeling for text generation, addressing performance gap between diffusion and autoregressive methods in language modeling. Sahoo et al. (2024) demonstrate that well engineered masked diffusion can achieve state-of-the-art results among diffusion models, approaching autoregressive perplexity within 15-25% on standard benchmarks.

### B.3.1 Initial architecture and Interesting points on choosing MDLM

The authors present a substitution-based way to parameterize the reverse diffusion process with two key features. First, it has zero masking probabilities, which means the model never predicts [MASK] tokens. Second, it includes carry-over unmasking, meaning that unmasked tokens stay the same during reverse diffusion. This approach simplifies the training objective analytically.

The model provides a simpler continuous-time variational lower bound that effectively reduces training variance using a low-discrepancy sampler (as discussed in Section 3.2 and Appendix D.3 in Sahoo et al. (2024)) and enhances numerical stability. The resulting objective is a weighted average of masked language modeling (MLM) losses, which shows a clear connection between diffusion models and BERT-style encoders.

Unlike earlier diffusion methods that only worked with fixed-length sequences, MDLM allows the generation of text of any length using a new semi-autoregressive decoding algorithm. This method employs previously generated tokens as prefixes for the next generation rounds, making it possible for the model to create longer texts gradually.

The framework uses a Diffusion Transformer (DiT) architecture that adds timestep conditioning into encoder-only transformers with rotary positional embeddings. One important finding is that leaving out explicit time dependence in the denoising network works well, while also letting the model optimize caching, leading to a 2x speedup in inference.

Most importantly, the authors show that modern engineering techniques, such as effective tokenization, stable numerical implementations, and updated architectures, greatly enhance performance, even for methods that were previously dismissed. This suggests that some performance gaps in earlier studies may have been due to implementation issues rather than fundamental flaws in the algorithms.

### B.3.2 Image conditioning

The MDLM framework's connection to BERT-style architectures and its sequential generation paradigm make it particularly well-suited for multimodal generation applications. The encoder-only design is suitable to incorporate visual features through cross-attention mechanisms, while the iterative process facilitates image-conditioned text generation that maintains long-range coherence.

To achieve this, we have gone further implementing a new methodology using the CLIP visual extractor as a feature extractor, along with cross-attention integration as an extension of the initial architecture.

Our approach incorporates image conditioning into DiT through a dedicated cross-attention mechanism, added to dynamic transformer blocks, enabling direct interaction between textual representations and visual features while preserving the original temporal conditioning mechanism.

We employ a dual-pathway image conditioning architecture that processes visual information using two complementary mechanisms. First, we utilize global visual conditioning via feature fusion, where spatially pooled image features are added to temporal embeddings and propagated through an Adaptive Layer Normalization (AdaLN) across all transformer layers. After that, we introduce spatial aware cross-attention in selected layers, where individual text tokens can attend to specific spatial patches from the image encoder's patch token representation.

This approach is interesting due the multi-granularity conditioning and the complementary information flow it provides. The architecture processes visual information at two different levels of details at the same time (global context through AdaLN and local spatial details one using cross-attention), which allows the models to have a better scene understanding. Additionally, the global pathway ensures that all text generation in influenced by visual context.

To further extend the image conditioning paradigm, we also studied the benefits of using two pre-trained text-only models as a backbone: the first one, with 40k steps due to hardware constraints, using Fineweb-Edu Lozhkov et al. (2024) (1.3T tokens) with a CLIP tokenizer; and the second one, consisting of OpenWebText ope for 1M training steps to the Huggingface hub Gokaslan et al. (2019) with a GPT2 tokenizer. This choice leads to faster convergence, improved text generation quality, and more stable training. Following this, we implemented a fine-tuning strategy with a modified DiT architecture, trained on both the CC3M dataset Sharma et al. (2018), which contains a training set of approximately 3.3 million image-caption pairs, and COCO 2017 Lin et al. (2015a;b).

### B.3.3 TEXT GENERATION (IMAGE CAPTIONING)

Repeating the idea expressed in subsection 4.2, we evaluate the text generation capabilities by producing captions conditioned on 1k real images selected randomly from the COCO2014 validation set and comparing the similarity between real and synthetic captions. In this case, the resulting mean cosine similarity is 0.50 (with a standard deviation of 0.14), indicating a moderate overall alignment with the real captions.

## C CROSS GUIDANCE

As mentioned in Section 3.2, we explore intrinsic alternatives to Classifier Free Guidance within the multimodal framework. We leverage the fact that textual prompts belong to the target distribution approximated by our model. In addition to learning probability paths interpolating between $Z_0 = (I_0, C_0)$ and $Z_1 = (\text{image, caption})$, during training we initialize the model with some probability $p$ ($p = 0.1$) with $\tilde{Z}_0 = (I_0, \text{caption})$.

Say the dataset $\mathcal{D} = (\cup i, \cup c)$ of image captions is the support of $p_1$, and that the initial distribution $p_0$ has support $\mathcal{D}_0 = (\cup i_0, \cup c_0)$. By training in the aforementioned way, the model learns paths interpolating between $\mathcal{D}_0 \cup (\cup c)$ and $\mathcal{D}$. Thus, when initializing the model with a prompt during sampling, this leads to guided generation. Crucially, prompt is allowed to change during generation within the data manifold learned by the model, which we believe allows for more diverse guided generation than CFG and mitigates the phenomenon of mode collapse.

### C.1 MODE COLLAPSE EXPERIMENT

To put these ideas to test, we design the following experiment.

We let $q_1^1$ be six gaussian distributions with means placed on a circle and variance of 0.5. This is our continuous distribution. Then, a grid is fixed to the plane, and for each given point $x$ we take the indices $i, j$ of the box containing it. This is our discrete distribution $q_1^2$. In order to mimic the fact that in image-caption datasets there are semantic concepts that are overrepresented, while others are underrepresented, we make our captions be $c = i + j$. This means labels along the top-right, bottom-left diagonal are underrepresented. See in Figure 6 a batch consisting of points and captions sampled from $p_1$.

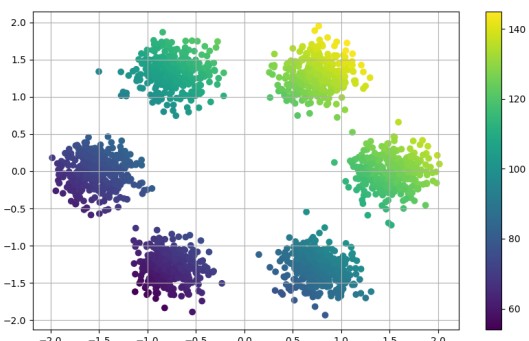

Figure 6: A batch of points and captions sampled from the target distribution of our experiment.

This dataset is interesting, because labels corresponding to central values $80 - 120$ are produced by more $i, j$ combinations. If we create a heatmap of cells in the grid colored by the amount of points in a batch that have a label $c$ such that $i + j = c$ we obtain a top-left-bottom-right distribution, see 7.

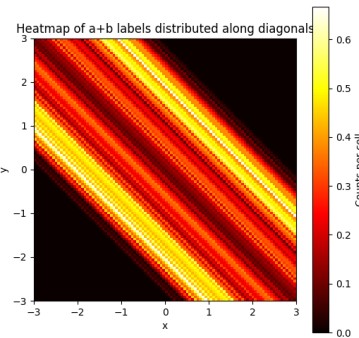

Figure 7: Cells in the grid colored by the amount of points in a batch that have a label $c$ such that $i + j = c$.

We now train a continuous flow on $q_1^1$, conditioning directly on the classes $c$ for $75k$ steps. We then sample using a CFG. On the other hand, we train a Cross-Modal flow, consisting on both a continuous flow approximating the path to $q_1^1$ and a discrete flow approximating the path to $q_1^2$. We sample in the way described above. In this case, the label we condition on is allowed to change during the generation. Results can be seen in 9. We use a CFG scale of $3.0$ in the first row, and of $1.5$ in the third row. We observe that CFG pushes points along the top-left-bottom-right diagonal, which is the area of labels that appear with higher density during training. This does not occur in the multimodal flow, but the generation is more noisy. Finally, the third row represents the best of both worlds.

## C.2 IMAGE INPAINTING

A great advantage of Cross-Guidance is that it allows us to perform controlled downstream tasks without any modifications to the model. One example is image inpainting, where we select an image and modify it by cutting out a section. One way to inpaint with a regular Flow Matching model would be to take the modified image, solve the ODE of our model in reverse time to get starting noise, and to then solve it in forward time hoping the model will reconstruct something

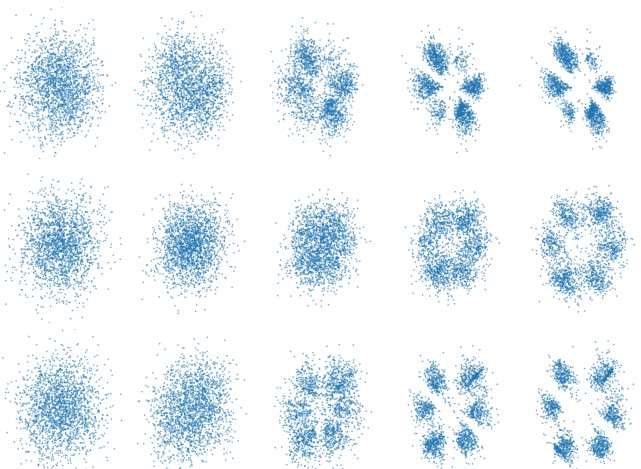

Figure 8: Experiment illustrating the phenomenon of mode collapse. The target distribution corresponds to six gaussians, each point assigned a label that corresponds to its position on a grid, i.e.: x and y indices of the box that contains it added together, which implies labels along the top-right, bottom-left diagonal are underrepresented. When sampling using all labels, CFG (first row), misrepresents those regions. CrossFlows (second row) doesn't, but leads to more noisy predictions. Third row represents CrossFlows + CFG.

close to the original image. But this method is not guaranteed to work, given that modified images are out-of-distribution.

In the multimodal setting, however, we have the evolution of the discrete modality to balance the forward process from the out-of-distribution input. The procedure is as follows:

1. Given a pair image/prompt, $(I_1, C_1)$, modify the image to obtain $\tilde{I}_1$.

2. Solve the ODE in backward time to obtain the initial noise of the modified image $\tilde{I}_0$.

3. Solve the ODE in forward time using $(\tilde{I}_0, C_1)$ as starting condition.

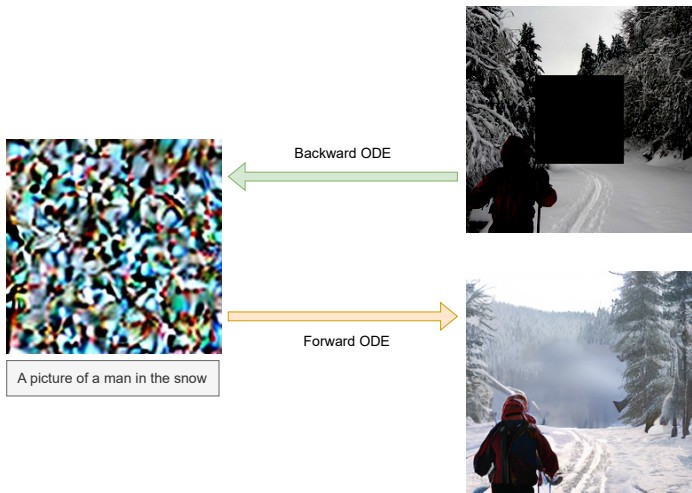

Figure 9: Image inpainting procedure with CrossFlows. We use Cross Guidance to guide generation from the noise obtained from computing the ODE in reverse time from the modified image.

