# OpenReview forum: "Cross-Modal Flows for Multimodal Generation"
_ICLR.cc/2026/Conference — ICLR 2026 Conference Withdrawn Submission_

### Official Review · Reviewer_Ntp5 · 2025-10-25

**Soundness:** 3
**Presentation:** 1
**Contribution:** 1
**Rating:** 2
**Confidence:** 4

**Summary:**

The paper proposes CrossFlows, a multimodal generative framework that jointly models a continuous flow (images) and a discrete flow (text) on the product space, rather than mapping modalities into a shared latent. It factorizes a conditional probability path into modality-specific components and trains interdependent generators so that each modality’s evolution influences the other, aiming for coupled cross-modal generation grounded in the Generator Matching formulation (linear combination of generators generates the probability path on the joint multimodal space). The method further introduces cross-guidance, which conditions generation by letting the guiding modality evolve on-manifold instead of fixing it, unlike classifier-free guidance. Experiments on COCO focus mainly on T2I and captioning, while at this scale the multimodal setup does not surpass a strong flow-matching baseline and the authors acknowledge small-scale training and limited comparisons as constraints.

**Strengths:**

- The method is explicitly tied to Generator Matching, using the linearity of generators to justify a factorized joint process over discrete (text) and continuous (image) space, which is a conceptually grounded way to couple modalities without a shared latent space.
- The paper attempts an on-manifold cross-guidance alternative to fixed-prompt CFG.

**Weaknesses:**

- **Unclear contribution:** The central novelty is limited. In essence, the method factorizes the probability path and trains two flows (continuous + discrete) that are linearly combined per Generator Matching; this seems principled but straightforward composition of known pieces rather than a new algorithmic proposal.
- **Insufficient evaluation and weak baselines:** Experiments are limited to COCO-scale text-image pairs; the multimodal setup underperforms a flow-matching image baseline on FID, and even with a pretrained T2I initialization it does not improve.
- **Narrow “multimodality”:** Despite claims of general multimodal generation, only text–image pair setting is demonstrated. No experiments on audio-text, video-text, or continuous–continuous space pairs (e.g., video–audio) are provided to validate the generality of cross-modal flows beyond T2I/I2T.
- **Limited evidence for cross-guidance effect:** Quantitative evidence on real datasets (e.g., diversity vs. fidelity trade-offs vs. CFG) is missing.
- **Presentation quality:** Key mechanisms (e.g., the exact conditioning interface between flows, stability/consistency of the coupled evolution, and cross-guidance behavior beyond the toy example) are described at a high level, and Figure explanations do not fully clarify design choices, making the overall method hard to understand.

**Questions:**

See weaknesses

---

### Official Review · Reviewer_iheS · 2025-10-29

**Soundness:** 2
**Presentation:** 1
**Contribution:** 1
**Rating:** 2
**Confidence:** 5

**Summary:**

The paper proposes CrossFlow, a multimodal flow-matching framework for joint text-image generation, using continuous flow matching for images and discrete flow matching for text captions. The author trained the model on MSCOCO to verify the algorithm's effectiveness. Additionally, representation alignment techniques such as REPA are incorporated to accelerate the multimodal flow training process.

**Strengths:**

- The paper approaches an important task, which is the multimodal joint generation of multiple modalities in a unified model.
- The paper discusses an interesting guidance scheme.

**Weaknesses:**

- **Bad presentation**

The paper consistently cites other papers in an inappropriate format, rendering it unreadable. For example, many citations are supposed to be wrapped in parentheses with \citep rather than being directly inserted in the paragraph with \citet.  Moreover, the word usage is very informal. The writing needs to be greatly improved in order to reach the standard of ICLR conference.

- **Missing many related, important works and leaving highly relevant methods undiscussed**.

Joint multimodal generation of text and images has been explored in many prior published works. To list a few Unidiffuser [1] in ICML 2023 considered joint generation of text and images with continuous diffusion, Diffuse Everything [2] in ICML 2025 considered joint generation of text and images with a combined discrete and continuous diffusion, and OmniFlow [3] in CVPR 2025 generalizes the setting to the multimodal generation of text, image and audio. These works all support any-to-any generation within a single model, and the adopted approach in CrossFlow closely mimics Diffuse-Everything [2] in its mathematical formulation.  Yet none of these works are discussed or compared in the paper, which makes the presented results less convincing. The author needs to discuss, cite these aforementioned prior works in the literature clearly and clarify the contributions of this paper

- **Missing experimental benchmark**

As is briefly mentioned in the previous comments, there is a lack of a convincing comparison in the experiment section to showcase the effectiveness of the proposed method. While I am fine with the small scale of the experiments, reproducing prior methods in the same setting and comparing them would greatly strengthen the work.

- **Technical errors**

The paper also contains some technical errors/confusions. For example, in Line 237, to correctly learn u^2 (which is the discrete velocity), a more appropriate choice for the divergence should be cross entropy or generalized KL instead of L^2, as per discrete flow matching or generation matching says. It's also mentioned later in Line 376 that the text model's training uses cross-entropy/generalized KL. The mismatch in the same paper would greatly increase the reader's confusion about the technical approaches.


**References**

[1] Bao, Fan, et al. "One transformer fits all distributions in multi-modal diffusion at scale." ICML 2023.

[2] Rojas, Kevin, et al. "Diffuse Everything: Multimodal Diffusion Models on Arbitrary State Spaces." ICML 2025

[3] Li, Shufan, et al. "Omniflow: Any-to-any generation with multi-modal rectified flows." CVPR 2025

**Questions:**

See the weaknesses section.

---

### Official Review · Reviewer_S5js · 2025-11-01

**Soundness:** 1
**Presentation:** 1
**Contribution:** 1
**Rating:** 2
**Confidence:** 3

**Summary:**

This paper develops CrossFlow for multimodal generation, learning flow on a joint discrete and continuous space. CrossFlow can perform text-to-image, image-to-text, and single-modality generation tasks.

**Strengths:**

The paper is trying to develop an image-language multimodal generative model by building upon the Generator Matching framework.

**Weaknesses:**

- Insufficient quantitative results: Table 1 only compares Cross-Modal with one existing baseline model on a single evaluation dataset, which is far from a valid comparison. Five metrics are involved, but 3 of them were missing for the baseline model.
- Low image quality in qualitative results: If zoomed in, the details in selected images are messed up. Additionally, the images produced are of significantly lower quality than those generated by existing multimodal generative models.
- It's hard to connect the motivation of the paper and its proposed approach (i.e., how the proposed method helps address the issues that are discussed in lines 28 to 65 in the Introduction section).

Without compelling experimental results, the novelty of the paper lacks justification, as numerous good performers are already available.

**Questions:**

Please refer to Weaknesses.

---

### Official Review · Reviewer_nmPL · 2025-11-01

**Soundness:** 4
**Presentation:** 3
**Contribution:** 1
**Rating:** 2
**Confidence:** 5

**Summary:**

The paper introduces a method for generating multimodal data using flow methods, taking from generator matching. Their method involves joining a flow model with a discrete flow model to generate (image, text) pairs. This allows for a joint generative process that creates the pair simultaneously, and their experiments demonstrate that the generated samples have a strong correlation.

**Strengths:**

- The paper is well-explained and easy to follow
- The method is well-based on mathematical foundations from the generator matching paper, which allows for precise generation up to learning error
- The paper demonstrates a technique for applying guidance for controllable generation by introducing pieces of information that are known

**Weaknesses:**

- The proposed methodology doesn't include conditional generation. By using cross-guidance it could be possible to have conditional generation. However, there is a discrepancy between the inputs that would be used and those used during training.

For instance, when generating an image conditioned on text. During training, the inputs are a noisy image and a noisy text, but during inference, it would be a noisy image with complete text, creating a significant discrepancy

- The paper has missed an important related work that targets the same problem at a similar scale Diffuse-Everything [1] which handles the joint generation, as well as the conditional generation (a feature missing in Cross-modal flow) of either modality. They do also train on text+image by combining a continuous diffusion with discrete diffusion, which is the main idea of the paper.

- CFG requires mixing two predictions, however cross-guidance seems to leverage just a single prediction by replacing some inputs with the known condition. Is this accurate, or is this prediction actually combined with the unconditional prediction?

- The related works in this paper is missing many important works. Although the authors mention that the point of the paper is to demonstrate the potential of flow models for multimodal data, it is important to discuss other possible approaches. Other than [1] some relevant works include Omniflow [2], JanusFlow [3], JetFormer [4], Show-O [5], UniD3 [6], and VersatileDiffusion [7].

[1] Rojas, Kevin, et al. "Diffuse Everything: Multimodal Diffusion Models on Arbitrary State Spaces." arXiv preprint arXiv:2506.07903 (2025).


[2] Li, Shufan, et al. "Omniflow: Any-to-any generation with multi-modal rectified flows." Proceedings of the Computer Vision and Pattern Recognition Conference. 2025.

[3] Ma, Yiyang, et al. "Janusflow: Harmonizing autoregression and rectified flow for unified multimodal understanding and generation." Proceedings of the Computer Vision and Pattern Recognition Conference. 2025.

[4] Tschannen, Michael, André Susano Pinto, and Alexander Kolesnikov. "Jetformer: An autoregressive generative model of raw images and text." arXiv preprint arXiv:2411.19722 (2024).

[5] Xie, Jinheng, et al. "Show-o: One single transformer to unify multimodal understanding and generation." arXiv preprint arXiv:2408.12528 (2024).

[6] Hu, Minghui, et al. "Unified discrete diffusion for simultaneous vision-language generation." arXiv preprint arXiv:2211.14842 (2022).

[7] Xu, Xingqian, et al. "Versatile diffusion: Text, images and variations all in one diffusion model." Proceedings of the IEEE/CVF international conference on computer vision. 2023.

**Questions:**

- The authors mention that for the discrete component, they use either masked or uniform as the source distribution, but in the experiments, it is not specified which one was used
- Clip embeddings are applied to text; however, the text is in a noisy state. If using masked diffusion, how are the masked inputs dealt with when applied to the clip embedder?

---

### Note · Authors · 2025-11-12

**Comment:**

In light of the reviews, and existence of similar works, we withdraw the paper.

**Withdrawal Confirmation:**

I have read and agree with the venue's withdrawal policy on behalf of myself and my co-authors.